# Hydrodeoxygenation of Pyrolysis Oil in Supercritical Ethanol with Formic Acid as an In Situ Hydrogen Source over NiMoW Catalysts Supported on Different Materials

**Mingyuan Zhang** [1,†]**, Xue Han** [2,†] **, Huanang Wang** [1] **, Yimin Zeng** [2,*] **and Chunbao Charles Xu** [1,*]

[1] Department of Chemical and Biochemical Engineering, Western University, London, ON N6A 3K7, Canada; mzhan582@uwo.ca (M.Z.); hwan482@uwo.ca (H.W.)
[2] CanmetMATERIALS, Natural Resources Canada, Hamilton, ON L8P 0A5, Canada; xue.han@nrcan-rncan.gc.ca
* Correspondence: yimin.zeng@nrcan-rncan.gc.ca (Y.Z.); cxu6@uwo.ca (C.C.X.)
† These authors contribute equally to this work.

**Abstract:** Hydrodeoxygenation (HDO) is one of the most promising approaches to upgrading pyrolysis oils, but this process normally operates over expensive noble metal catalysts (e.g., Ru/C, Pt/Al$_2$O$_3$) under high-pressure hydrogen gas, which raises processing costs and safety concerns. In this study, a wood-derived pyrolysis oil was upgraded in supercritical ethanol using formic acid as an in situ hydrogen source at 300 °C and 350 °C, over a series of nickel–molybdenum-tungsten (NiMoW) catalysts supported on different materials, including Al$_2$O$_3$, activated carbon, sawdust carbon, and multiwalled nanotubes (MWNTs). The upgrading was also conducted under hydrogen gas (an ex situ hydrogen source) for comparison. The upgrading process was evaluated by oil yield, degree of deoxygenation (DOD), and oil qualities. The NiMoW/MWNT catalyst showed the best HDO performance among all the catalysts tested at 350 °C, with 74.8% and 70.9% of oxygen in the raw pyrolysis oil removed under in situ and ex situ hydrogen source conditions, respectively, which is likely owing to the large pore size and volume of the MWNT support material, while the in situ hydrogen source outperformed the ex situ hydrogen source in terms of upgraded oil yields and qualities, regardless of the catalysts employed.

**Keywords:** pyrolysis oil; catalytic hydrodeoxygenation upgrading; supercritical ethanol; in situ hydrogen source; NiMoW catalysts; different supports

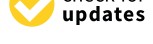



## 1. Introduction

It is of utmost importance to substitute fossil fuels with renewable energy when considering the growing world population, greenhouse gas (GHG) emissions, and the depletion of fossil fuels [1]. Therefore, the utilization of biomass and biofuels has attracted increasing attention worldwide due to their carbon neutrality and renewability [2–4]. Lignocellulosic biomass, such as forestry/agricultural residues composed of cellulose, hemicellulose, and lignin, is one of the most attractive biomass feedstocks for the production of biofuels or chemicals, as it is abundant, widely available, and low-cost [5–7].

The dominant—and currently the only—commercially utilized method to transform dry biomass into liquid fuels is through fast pyrolysis. [8]. It operates commonly at 300–700 °C with a 10–200 °C/s heating rate and 0.5–10 s residence time to produce bio-oil, combustible gases, and biochar [9]. However, the low-grade qualities of pyrolysis oil, e.g., thermal instability, high water and oxygen contents, and low heating value, are the main challenges to utilizing biofuels as "drop-in" fuels directly [10]. Different methods have been used to improve bio-oil quality. These include emulsification [11], hydrodeoxygenation (HDO) [12], steam reforming [13], catalytic cracking [14], and supercritical fluid treatment [15].

HDO is a promising method to improve bio-oil quality by removing oxygen atoms at 300–450 °C, usually in a hydrogen donor solvent such as tetralin and under high hydrogen gas pressure (up to 200 bar) [16–18]. Pressurized hydrogen gas is an external or ex situ hydrogen source, but economic and safety issues are obstacles to produce upgraded oil using pressurized hydrogen gas at industrial scale. In situ hydrogen sources, e.g., formic acid, have been studied as substitutes. Formic acid, obtained from bioresources, is regarded as a sustainable and safer in situ hydrogen source for bio-oil upgrading [19].

Instead of conventional hydrogen donor solvents, supercritical fluids have been employed in the HDO of bio-oils, which has become a hotspot in recent years [20,21]. A supercritical fluid acts as a suitable reaction medium due to its liquid-like solubility and heat transfer rate, gas-like diffusivity, and high miscibility to both liquid and gas, hence creating a homogeneous phase in the reactor [22]. Supercritical ethanol (Tc: 241 °C, Pc: 63 bar) can also be a hydrogen-donating solvent, and can reduce the feeding viscosity of pyrolysis bio-oil in a continuous flow reactor [23].

In an HDO process, catalysts play a significant role in the deoxygenation efficacy. In general, the catalysts applied for HDO treatment can be classified into two categories in terms of active phases: transitional metals and noble metals. Bjelić et al. [24] compared the effects of different transition metal catalysts (Cu/C, Ni/C) with noble metal catalysts (Pd/C, Pt/C, Rh/C, Ru/C) on the HDO of aromatic lignin monomers. It was found that the noble metal catalysts, especially the Ru/C catalyst, exhibited better HDO turnover performance than the other catalysts. In a study by Oh et al. [25], Ru/C and Pt/C were used in the hydrodeoxygenation of bio-oil at 250, 300, and 350 °C; it was found that the degree of deoxygenation can reach to 73.7% over the Pt/C catalyst at 350 °C. It has been widely reported that noble metal catalysts exhibit high catalytic activities in the HDO treatment of bio-oils [26–29]. The application of noble metal catalysts is, however, limited for industrial-scale applications due to their extremely high costs [30].

Inexpensive catalysts, such as conventional hydrotreatment or petroleum refinery catalysts, including NiMoW, NiMo, and CoMo sulfides supported on different materials, have been applied for bio-oil catalytic HDO upgrading [31]. Moreover, trimetallic catalysts show better performance than bimetallic catalysts [32]; therefore, this work selects NiMoW as the target metal, in which Mo and W are the active phase and Ni is the promoter. In addition, the catalyst supports were found to play an important role in determining the catalytic activity, partly due to their different levels of acidity [33]. In this work, NiMoW on different supports (alumina, sawdust carbon, activated carbon, multiple wall nanotubes) was investigated on the HDO upgrading of pyrolysis oil in supercritical ethanol at 300 and 350 °C, with formic acid as an in situ hydrogen source. Catalytic performance was evaluated by product yields and upgraded oil quality.

## 2. Experimental

### 2.1. Materials

The crude bio-oil used in this work was a commercially available fast pyrolysis oil derived from woody biomass, whose compositions and properties are shown in Table 1. The commercial NiMoW/$Al_2O_3$ catalyst, consisting of 10% Ni, 6% Mo, 5%W, was obtained from Fushun Petrochemical Research Institute of China Petrochemical Corp. The other catalysts used in this work were self-prepared. Nickel(II) nitrate hexahydrate (>98%), ammonium molybdate tetrahydrate (>98%), ammonium tungsten oxide hydrate (>98%), acetone, and formic acid (88%) were ACS reagent-grade chemicals purchased from Sigma Aldrich. Activated carbon pellet (AC) and multiwalled carbon nanotubes (MWNTs) were purchased from Fisher scientific and Cheap Tubes Inc. (Grafton, VT, USA), respectively. Pinewood sawdust, used as the precursor for sawdust carbon (SC), was supplied by a sawmill in South Ontario, whose proximate and ultimate analyses are listed in Table 2. Commercial Alcohols provided anhydrous ethanol (density 0.7885 g/mL, water content < 0.1 wt.%).

**Table 1.** Compositions and properties of the wood-derived pyrolysis oil.

| Total Acid Number (TAN) (mgKOH/g) | Viscosity (mPa·s) | Elemental Analysis [a] | | | | | H/C (−) | O/C (−) | HHV [c] (MJ/kg) |
|---|---|---|---|---|---|---|---|---|---|
| | | C (wt.%) | H (wt.%) | O [b] (wt.%) | N (wt.%) | S (wt.%) | | | |
| 136.0 | 105.3 | 46.21 | 7.11 | 46.45 | 0.21 | 0.02 | 1.85 | 0.75 | 17.49 |

[a] On dry basis; [b] determined by difference assuming negligible ash; [c] higher heating value (HHV), calculated by the Dulong equation: HHV (MJ/kg) = 0.338%C + 1.428(%H − %O/8) + 0.095%S.

**Table 2.** Proximate and ultimate analysis of the pinewood sawdust.

| Proximate Analysis (wt.%) [a] | |
|---|---|
| Volatile matter | 77.62 |
| Ash | 0.44 |
| Fixed carbon | 21.94 |
| **Elemental analysis (wt.%) [a]** | |
| C | 50.65 |
| H | 5.47 |
| O [b] | 43.35 |
| N | 0.09 |
| S | - |

[a] On dry basis; [b] determined by difference (100%—C%—H%—N%—S%—ash%).

### 2.2. Synthesis of Catalysts

NiMoW/AC, NiMoW/MWNTs, and NiMoW/SC catalysts were self-prepared using activated carbon pellets (AC), multiwalled carbon nanotubes (MWNTs), and pinewood sawdust, respectively, as the support materials, via incipient wetness impregnation method [34].

The procedure of the method is briefly described below: (1) Impregnating 8 g support material with 4.99 g nickel(II) nitrate hexahydrate, 1.12 g ammonium molybdate tetrahydrate, and 0.68 g ammonium tungsten oxide hydrate for the targeted metal loadings: 10.0 wt.% Ni, 6 wt.% Mo, 5 wt.% W. The impregnation was conducted at room temperature for 10 min with manual stirring when the solution was completely adsorbed by the support. (2) Drying the metal-impregnated support at 120 °C overnight in air. (3) Calcining/carbonizing the dried metal-impregnated support in a tubular furnace in $N_2$ flow at 400 °C for 3 h.

### 2.3. HDO Experiments

Bio-oil HDO upgrading experiments were conducted in a 100 mL or 500 mL stainless-steel high-pressure autoclave reactor at 300 °C and 350 °C, respectively. All experiments were repeated 2–3 times to ensure reproducibility and statistical significance of the experimental results. For a typical operation using the 100 mL autoclave reactor, 20 g pyrolysis bio-oil, 20 g ethanol, 1.6 g catalyst, and 2.73 g formic acid were charged into the reactor. In the experiments with the 500 mL reactor, 40 g pyrolysis bio-oil, 40 g ethanol, 3.2 g catalyst, and 5.46 g were loaded into the reactor. Catalysts used in HDO reaction included NiMoW/$Al_2O_3$, NiMoW/SC, NiMoW/AC, and NiMoW/MWNTs. The reactor was sealed and leak tested with compressed nitrogen, and then the residual air inside the reactor was removed by nitrogen purging and releasing 3 times. Finally, the reactor was pressurized by 35 bar nitrogen gas (or hydrogen when using pressurized hydrogen as the ex situ hydrogen source), and heated to a desired temperature (300, 350 °C) under constant stirring (~120 rpm) for 2 h.

At the end of each experiment, the reactor was quenched in a water bath. After the reactor was cooled to ambient temperature (~25 °C), the gaseous products were collected into a gas bag and analyzed by Micro GC to determine the gaseous product composition and yield. Subsequently, the reactor was opened, and the reaction mixture was moved into a 500 mL beaker. The reactor and stirrer were washed with acetone thrice, and the collected washings were blended with the reaction mixture. The resulting mixture of

reaction product and solvent was then filtrated under vacuum. After oven drying at 105 °C for 12 h, the solid product retained on the filter paper included solid residue (carbon/coke) and the spent catalyst (whose mass was assumed constant during the experiments). The filtrate was evaporated in a preweighed flask under reduced pressure to separate solvents and water from upgraded bio-oil, when the pyrolytic water and low-boiling point volatiles were lost in the evaporation.

The yields of products were calculated as follows:

$$Yield\ of\ upgraded\ bio\ oil\ (wt.\%) = \frac{Mass\ of\ upgraded\ bio\ oil}{Dry\ mass\ of\ bio\ crude\ oil} \times 100\% \tag{1}$$

$$Yield\ of\ solid\ residue\ (wt.\%) = \frac{Mass\ of\ solid\ residue}{Dry\ mass\ of\ bio\ crude\ oil} \times 100\% \tag{2}$$

$$Yield\ of\ gas\ products\ (wt.\%) = \frac{Mass\ of\ gas\ products}{Dry\ mass\ of\ bio\ crude\ oil} \times 100\% \tag{3}$$

$$Yield\ of\ pyrolytic\ water\ and\ low\ boiling\ point\ volatiles\ (wt.\%) = 100 - yield\ of\ upgraded\ bio\ oil - yield\ of\ solid\ residue - yield\ of\ gas\ products \tag{4}$$

### 2.4. HDO Products Analysis

An Agilent Micro-GC 3000 equipped with a thermal conductivity detector (GC-TCD) was utilized for analyzing the gas composition.

A GC-MS (Agilent 7890A GC/MSD-5977) with an HP-5MS column (30 m × 0.25 mm × 0.25 μm) was utilized for qualitative identification of the primary volatile components in both the crude bio-oil and the upgraded bio-oil samples. Pure helium was utilized as the carrier gas, flowing at a rate of 1 mL/min. The GC oven temperature program consisted of holding the temperature at 45 °C for 5 min, followed by an increase to 250 °C with a ramping rate of 8 °C/min, and a final hold at 250 °C for 10 min. For sample preparation, the upgraded oil was diluted with acetone at a volume ratio of 1:30. The mixture was filtered through a 0.45 μm filter before injection. The Vario EL Cube elemental analyzer (Elementar, Germany) was used for determining the elemental compositions (C, H, N, and S). The higher heating value (HHV) in MJ/kg was calculated using the DuLong formula, as follows [35]:

$$HHV\ (MJ/kg) = 0.338C\% + 1.428\%(H\% - O\%/8) + 0.095S\% \tag{5}$$

Degree of deoxygenation (*DOD*) was calculated to evaluate the extent of oxygen removal in the HDO process by the following formula [36–38]:

$$DOD = 1 - \left( \frac{O(wt.\%)_{upgraded\ oil,\ dry\ base}}{O(wt.\%)_{crude\ bio\ oil,\ dry\ base}} \right) \times 100\% \tag{6}$$

The total acid number (TAN) was determined via titration following ASTM D664 [39]. Moreover, viscosities of the crude and upgraded bio-oils were characterized by Brookfield viscometer (Middleboro, MA, USA) at 50 °C. A PerkinElmer FTIR spectrometer (Waltham, MA, USA) was utilized for analyzing functional structure of the oils in the wavenumber range of 4000–600 $cm^{-1}$ at a resolution of 8 $cm^{-1}$.

### 2.5. Catalyst Characterization Methods

The actual metal contents of the synthesized catalysts were measured by inductively coupled plasma atomic emission spectroscopy (ICP-AES) (Varian (Agilent) Vista Pro Radial ICP-OES) [40]. The textural properties (BET specific surface area, average pore diameter, and total pore volume) of the fresh catalysts were characterized by $N_2$ adsorption–desorption at 77K using Quantachrome NOVA1200e [34]. Powder X-ray diffraction (XRD)

analysis was performed to examine the crystalline structure of the catalyst using Ni-filtered Cu-K$\alpha$ radiation on a Philips PW 1050-3710 diffractometer [41].

## 3. Results and Discussion

### 3.1. Fresh Catalyst Characterizations

The actual metal contents, BET specific surface area, average pore diameter, and total pore volume of the as-synthesized NiMoW catalysts supported on different materials are listed in Tables 3 and S14, and nitrogen adsorption/desorption isotherms of all NiMoW catalysts are depicted in Figure S1. Ni and Mo contents were detected and are listed; W could not be tested due to the limitation of the equipment. Ni and Mo tend to agglomerate due to the low surface area and pore volume, which leads to metal being lost during catalyst synthesis (impregnation → drying → calcination). Furthermore, the Ni and Mo contents were lower than the target loading amount, which could be improved by adding external mixing, using alternative metal precursors, or using a successive incipient wetness impregnation method to synthesize the catalyst. As clearly shown in this Table, the activated carbon-supported catalyst has the largest surface area (706.8 m$^2$/g), while the average pore diameter and total pore volume of the MWNT-supported catalyst is much higher than the other catalysts. A larger surface area can expose more active sites to the bio-oil, and a larger pore diameter and pore volume would improve the diffusion of bio-oil molecules into the inner pores, both of which would promote the bio-oil HDO reactions [42]. As depicted in Figure S1, NiMoW supported on sawdust shows a Type II isotherm, according to the International Union of Pure and Applied Chemistry (IUPAC) classification, suggesting a nonporous structure, while other NiMoW catalysts supported on Al$_2$O$_3$, activated carbon, multiwalled nanotubes show a Type IV isotherm, indicating the mesoporous structure of these catalysts.

**Table 3.** Actual metal contents and textural properties of NiMoW catalysts supported on different materials.

| | NiMoW/ Al$_2$O$_3$ | NiMoW/ AC | NiMoW/ SC | NiMoW/ MWNTs |
|---|---|---|---|---|
| Ni (wt.%) | 10 | 5.19 | 3.99 | 5.15 |
| Mo (wt.%) | 6 | 2.4 | 1.98 | 2.8 |
| W (wt.%) | 5 | - | - | - |
| Surface area (m$^2$/g) | 131.9 | 706.8 | 7.7 | 86.9 |
| Average pore diameter (nm) | 5.8 | 2.3 | 5.7 | 17.4 |
| Total pore volume (cm$^3$/g) | 0.19 | 0.41 | 0.011 | 0.72 |

The XRD patterns of the NiMoW catalysts are shown in Figure 1. Only $\gamma$-Al$_2$O$_3$ was detected on the NiMoW/Al$_2$O$_3$ catalyst, showing that Ni, Mo, and W have good dispersion on this commercial catalyst. NiMoW/AC has weak diffraction peaks of MoO$_3$ and NiO$_2$, indicating that Ni and Mo slightly aggregate on the catalyst surface and W may be highly dispersed on the support. Moreover, the (110) plane of C$_8$ was detected at 29°, which may be from activated carbon [43]. Weak peaks of Mo$_4$O$_{11}$ and WO$_3$, but no Ni signal, were detectable in the NiMoW/SC catalyst, suggesting good dispersion of Ni species in the catalyst. There is a sharp XRD peak of MoO$_2$ on the spectrum of the NiMoW/MWNTs catalyst, suggesting the presence of poorly dispersed MoO$_2$ species with a large crystalline size [44].

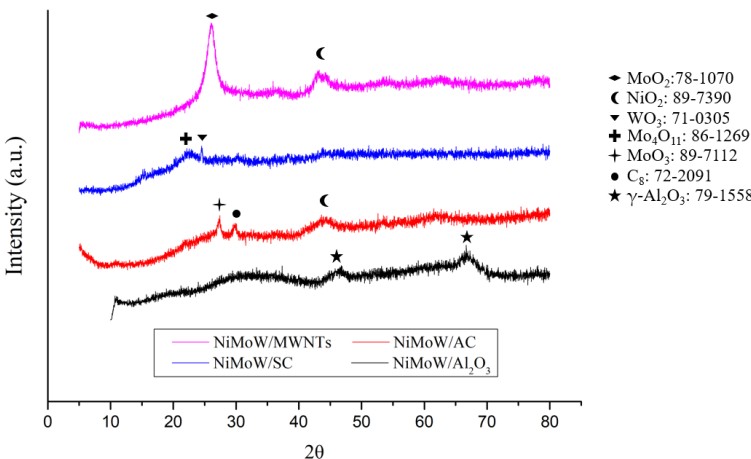

**Figure 1.** XRD patterns of the NiMoW catalysts supported on different materials.

### 3.2. HDO Products Distribution

The performance of the NiMoW catalysts supported on different materials in bio-oil HDO was studied under 35 bar initial pressure of $N_2$ or $H_2$ for a fixed reaction time of 2 h in supercritical ethanol at 300 or 350 °C with in situ (formic acid) or ex situ ($H_2$ gas) hydrogen sources. Figure 2 shows the product distribution in the catalytic HDO experiments.

In Figure 2a, the product yields of the blank experiments are displayed. These experiments involved the hydrodeoxygenation (HDO) of pyrolysis oil without the addition of a catalyst, formic acid, or ethanol. The tests were conducted at 300 °C, and when formic acid or ethanol was not included in the process, a NiMoW/MWNTs catalyst was used. Figure 2a illustrates that the absence of a catalyst results in a lower oil yield (24.3 wt.%) compared to most of the catalytic HDO experiments, while the solid residue yield (25.7 wt.%) shows minimal difference from those in the catalytic process, with a few exceptions. These findings suggest that the primary effect of the catalysts in this study is to enhance oil production. In the absence of formic acid in the process, a reduction in oil yield of about 19 wt.% and a rise in solid yield of 22 wt.% are observed, with no significant changes in the yields of gas products. When ethanol is not included in the process, the oil yield dramatically decreases to 10 wt.%, and the solid residue yield increases to 38.8 wt.%. These results suggest that supercritical ethanol provides a homogeneous reaction environment that enhances hydrogen solubility [45] and improves the effectiveness of formic acid. Additionally, ethanol itself can act as a hydrogen donor, which significantly restricts the formation of solid residues.

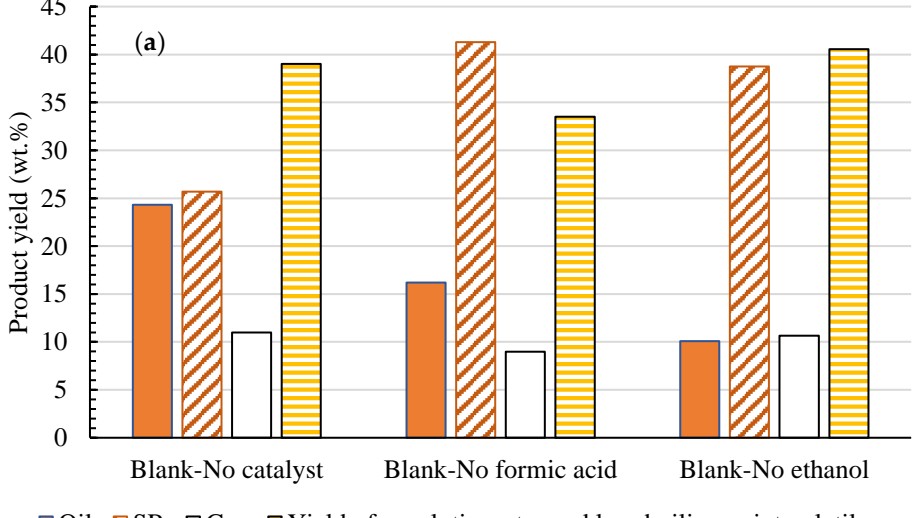

**Figure 2.** *Cont.*

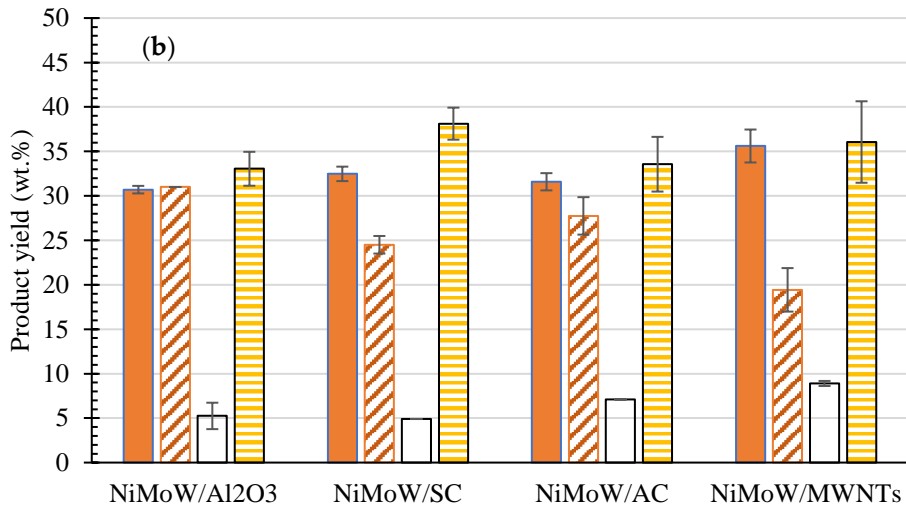

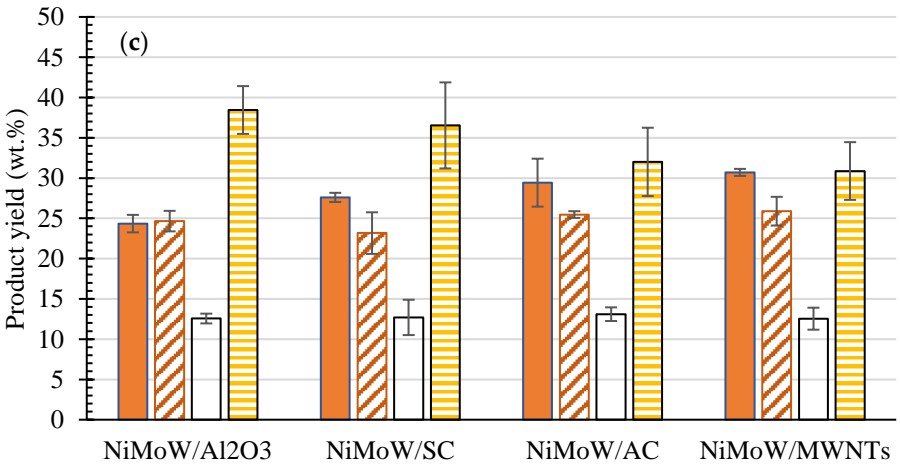

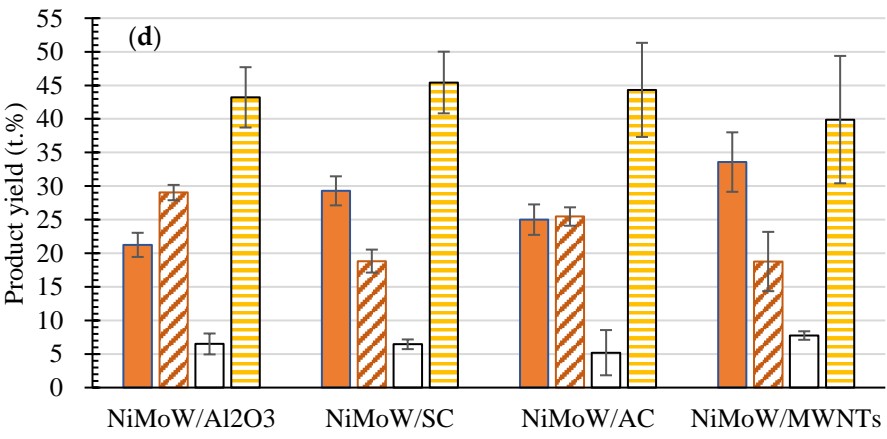

**Figure 2.** HDO Product distributionS of pyrolysis oil blank test at 300 °C (**a**), over NiMoW catalysts supported on different materials in supercritical ethanol with formic acid at 300 °C (**b**) and 350 °C (**c**), or with hydrogen gas at 350 °C (**d**).

As shown in Figure 2b, the upgraded oil yields were over 30 wt.% for all NiMoW catalysts in supercritical ethanol at 300 °C with the in situ hydrogen source, while the solid residue yield over the $NiMoW/Al_2O_3$ catalyst was 31.0 wt.%, higher than those over the carbon-supported catalysts, ranging from 19.4 to 27.8 wt.%. As reported by Gayubo et al. [46] and Valle et al. [47], the internal coke formation on the HDO catalysts was caused by depolymerization/condensation of reaction intermediates/products derived from bio-oil, which was catalyzed by the acidic sites. Thus, the reduced coke formation on carbon-supported catalysts could likely be attributed to their low acid sites [48]. Among all catalysts, NiMoW/MWNTs exhibited the best performance in terms of the highest upgraded oil yield (35.6 wt.%) and the lowest solid residue yield (19.4 wt.%), which could be owing to its well-developed pore structure. A larger specific surface area is advantageous, as it can disperse the active metals more efficiently and provide a greater number of active sites where catalytic reactions can occur. During the HDO reaction, reactant molecules are known to diffuse through the mesopores of the catalyst and adsorb onto the surface before the catalytic reaction takes place [49]. Despite having a relatively low specific surface area of 86.9 $m^2/g$, the mesoporous MWNT-supported catalysts demonstrate high HDO activity. This suggests that both the large specific area and mesoporous structure of the catalysts contribute to their superior catalytic performance owing to the better dispersion of metal particles, which prevents agglomeration. In addition, higher pore volume and larger pore diameter can improve the diffusion of reactant molecules and the dispersion of active sites in catalysts.

Figure 2c depicts the HDO product distribution in the tests in supercritical ethanol at 350 °C with the in situ hydrogen source. Compared with those presented in Figure 2b, the upgraded oil yields over all NiMoW catalysts decreased slightly, accompanied with slightly reduced solid residue yields but significantly increased gas yields when raising the reaction temperature from 300 °C to 350 °C. The reasons for the significantly increased gas yield at a high temperature could be due to enhanced cracking reactions at a higher temperature [50]. Similarly, among all catalysts, NiMoW/MWNTs produced the highest yield of upgraded oil.

The product distributions in the HDO experiments under a conventional ex situ hydrogen source ($H_2$ gas) at 350 °C are present in Figure 2d. The highest upgraded oil yield and lowest solid residue yield was also obtained over the NiMoW/MWNT catalyst, which suggests that MWNTs could be the best support for the upgraded oil yield. Moreover, all carbon-supported NiMoW catalysts produced higher yields of upgraded oil than the $Al_2O_3$-supported catalyst in HDO tests with either in situ or ex situ hydrogen sources at 300 °C or 350 °C. This is likely because the carbon-based catalysts have fewer and/or weaker acid sites than the alumina-supported catalyst, hence suppressing the condensation/repolymerization of the oil molecules/intermediates during the HDO process [51].

### 3.3. Upgraded Oil Properties

Table 4 compares the TAN, viscosity, and elemental compositions of the upgraded oils obtained from the HDO of pyrolysis oil over NiMoW catalysts supported on different materials with in situ or ex situ hydrogen sources at 300 °C or 350 °C. Compared with the properties of crude pyrolysis oil presented in Table 1, HDO upgrading, leading to a DOD of 47.0–74.8%, significantly improved the oil properties, decreasing TAN from 136.0 mgKOH/g to 47–69 mgKOH/g, viscosity from approx. 105 mPa·s to 13–32 mPa·s, and oxygen content from approx. 46 wt.% to 12–24 wt.%, while increasing HHV from 17.5 MJ/kg to 29–38 MJ/kg. It is worth mentioning that the upgraded oils obtained with the ex situ hydrogen source at 350 °C have an oxygen content even higher than those obtained with the in situ hydrogen source at 300 °C, suggesting the superior performance of formic acid, which could serve as a reactive hydrogen-donating solvent and promote HDO reactions through the hydrogen transfer hydrogenation mechanism [52]. This result is the opposite of that obtained in our previous study [53], where the ex situ hydrogen source outperformed the in situ hydrogen source in the HDO of pyrolysis oil at higher tem-

peratures (325 or 350 °C) over a noble metal catalyst (Ru/C) with respect to the upgraded oil quality and DOD. It is possible that formic acid completely decomposed over a noble metal catalyst at a high temperature, so it could not promote HDO reactions through the hydrogen transfer hydrogenation mechanism anymore [52,54]. From Table 4, the HDO upgrading over the NiMoW/MWNT catalyst with the in situ hydrogen source at 350 °C achieved 74.8% DOD, and obtained an upgraded oil with TAN as low as 47.2 mgKOH/g and HHV as high as 37.53 MJ/kg.

**Table 4.** Properties/compositions of upgraded oils obtained from HDO of pyrolysis oil at 300 °C or 350 °C for 2 h over NiMoW catalysts supported on different materials with in situ or ex situ hydrogen sources.

| Sample | TAN (mgKOH/g) | Viscosity (mPa·s) | C (wt.%) | H (wt.%) | N (wt.%) | S (wt.%) | O (wt.%) | HHV (MJ/Kg) | DOD (%) |
|---|---|---|---|---|---|---|---|---|---|
| In situ hydrogen source at 300 °C for 2 h | | | | | | | | | |
| NiMoW/Al$_2$O$_3$ | 52.4 | 23.4 | 75.66 | 9.76 | 0.38 | 0.00 | 14.20 | 36.98 | 69.27 |
| NiMoW/SC | 69.3 | 16.2 | 76.13 | 6.60 | 0.30 | 0.00 | 16.97 | 32.12 | 63.27 |
| NiMoW/AC | 56.1 | 16.7 | 77.34 | 5.96 | 0.23 | 0.00 | 16.47 | 31.71 | 64.36 |
| NiMoW/MWNTs | 68.8 | 18.5 | 77.17 | 5.41 | 0.29 | 0.00 | 17.13 | 30.76 | 62.93 |
| In situ hydrogen source at 350 °C for 2 h | | | | | | | | | |
| NiMoW/Al$_2$O$_3$ | 55.6 | 20.9 | 75.81 | 9.85 | 0.46 | 0.00 | 13.88 | 37.21 | 69.96 |
| NiMoW/SC | 59.6 | 25.1 | 76.01 | 8.03 | 0.31 | 0.00 | 15.65 | 34.36 | 66.13 |
| NiMoW/AC | 62.7 | 32 | 76.38 | 7.73 | 0.25 | 0.00 | 15.64 | 34.06 | 66.15 |
| NiMoW/MWNTs | 47.2 | 21.2 | 79.09 | 9.01 | 0.26 | 0.00 | 11.64 | 37.53 | 74.81 |
| Ex situ hydrogen source at 350 °C for 2 h | | | | | | | | | |
| NiMoW/Al$_2$O$_3$ | 56.40 | 13.60 | 75.13 | 5.60 | 0.34 | 0.00 | 18.93 | 30.01 | 59.04 |
| NiMoW/SC | 57.30 | 18.60 | 74.89 | 5.99 | 0.33 | 0.00 | 18.79 | 30.51 | 59.34 |
| NiMoW/AC | 65.50 | 13.00 | 68.00 | 7.30 | 0.21 | 0.00 | 24.49 | 29.04 | 47.01 |
| NiMoW/MWNTs | 59.10 | 20.60 | 78.46 | 7.81 | 0.28 | 0.00 | 13.45 | 35.27 | 70.89 |

A van Krevelen diagram is frequently utilized to evaluate the efficacy of HDO of bio-oils in terms of both deoxygenation and hydrogenation [38]. The van Krevelen diagram of the crude pyrolysis oil and upgraded oils obtained from HDO in supercritical ethanol at 300 °C or 350 °C for 2 h over NiMoW catalysts supported on different materials with in situ or ex situ hydrogen sources is illustrated in Figure 3. The HDO upgrading drastically decreased the O/C molar ratio from 0.75 for the crude pyrolysis oil to 0.11–0.27 for the upgraded oils, demonstrating effective hydrodeoxygenation in the HDO process, while the H/C molar ratio of the oil also dropped, likely due to the dehydration reactions occurring in the process. The van Krevelen diagram clearly shows that the in situ hydrogen source outperformed the ex situ hydrogen source with respect to both deoxygenation and hydrogenation of the bio-oil. In particular, HDO upgrading over the NiMoW/MWNTs catalyst with the in situ hydrogen source at 350 °C obtained an upgraded oil with the lowest O/C molar ratio (~0.1) and a relatively high H/C molar ratio (~1.4).

In summary, in terms of both yields and properties of the upgraded oils, the NiMoW catalyst supported on MWNTs outperformed those supported on other materials (Al$_2$O$_3$, AC and SC) in HDO of pyrolysis oil in supercritical ethanol with in situ or ex situ hydrogen sources, and the in situ hydrogen source outperformed the ex situ hydrogen source in the HDO upgrading over all catalysts.

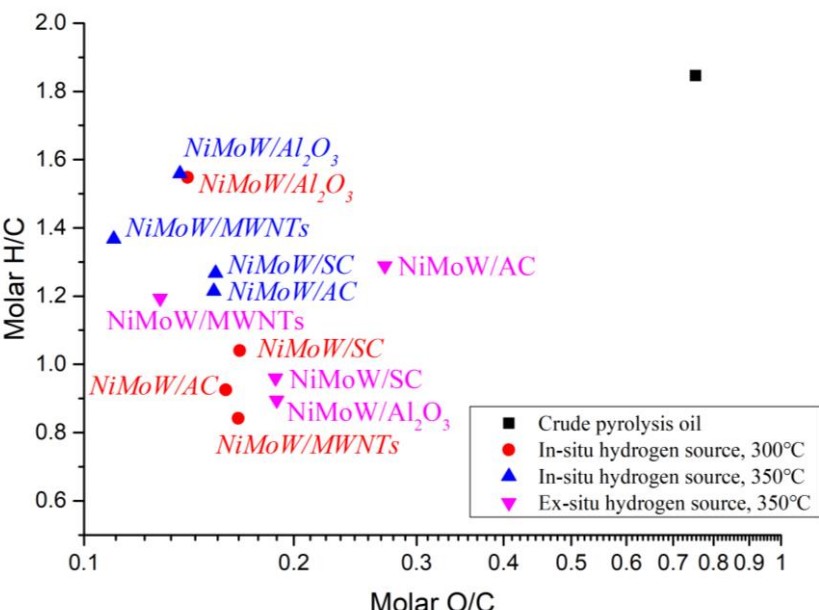

**Figure 3.** Van Krevelen diagram of the crude pyrolysis oil and upgraded oil obtained from HDO in supercritical ethanol at 300 °C or 350 °C for 2 h over NiMoW catalysts supported on different materials with in situ or ex situ hydrogen sources.

### 3.4. GC-MS Analysis of the Oils

The chemical compositions of the pyrolysis oil and upgraded oil obtained from HDO over NiMoW catalysts supported on different materials in supercritical ethanol with formic acid at 300 °C or 350 °C or with hydrogen gas at 350 °C were qualitatively analyzed by GC-MS and reported in peak area %, as shown in Figure 4. It should be noted that only volatile compounds were detectable by GC-MS. In this work, the volatile compounds are categorized into phenols, esters, ketones, aromatics, hydrocarbons, alcohols, aldehyde, acids, ethers, and others. As expected, the main compositions of the wood-derived pyrolysis oil mainly include oxygenated compounds, including phenols, ketones, and acids, derived from the decomposition of lignocellulosic biomass [55]. As clearly shown in the Figure, acids and aldehydes were eliminated, while hydrocarbons, aromatics, and esters content increased after the catalytic HDO upgrading. This suggests that acid compounds could be converted into esters or ketones by esterification (with the ethanol solvent), ketonization, hydrodeoxygenation, and hydrogenation reactions occurring during the HDO process [53], as evidenced by the drastic decreases in TAN and O/C molar ratio for the upgraded oils (Table 4 and Figure 3).

### 3.5. FT-IR Spectra of the Oils

The functional groups of the pyrolysis oil and upgraded oils were analyzed by FT-IR at a range of 4000–600 cm$^{-1}$. The FT-IR spectra of the crude bio-oil and upgraded oils obtained from HDO over different catalysts in supercritical ethanol with the in situ hydrogen source and over NiMoW/MWNTs with the ex situ hydrogen source at 350 °C are presented in Figure 5. The IR absorption peak between 3700–3000 cm$^{-1}$ can be ascribed to the hydroxyl group (−OH) in water, carboxylic acids, alcohols, and phenolics compounds. The peak at 1700–1710 cm$^{-1}$ represents the C=O stretching in acids and esters. Compared with the spectrum of the crude pyrolysis oil, the −OH and C=O bonds weaken, accompanied by an increased C=C aromatic bond (at approx. 1600 cm$^{-1}$), which suggests the removal of carboxylic acids and water and an increase in aromatics during the HDO process. Moreover, the stretching vibrations of the C-H group in hydrocarbons and aromatics appear clearly at 2960 cm$^{-1}$, 2925 cm$^{-1}$, and 2870 cm$^{-1}$. Thus, FTIR analysis further confirms the improved oil properties by HDO upgrading, supporting the results of other analyses, as discussed previously.

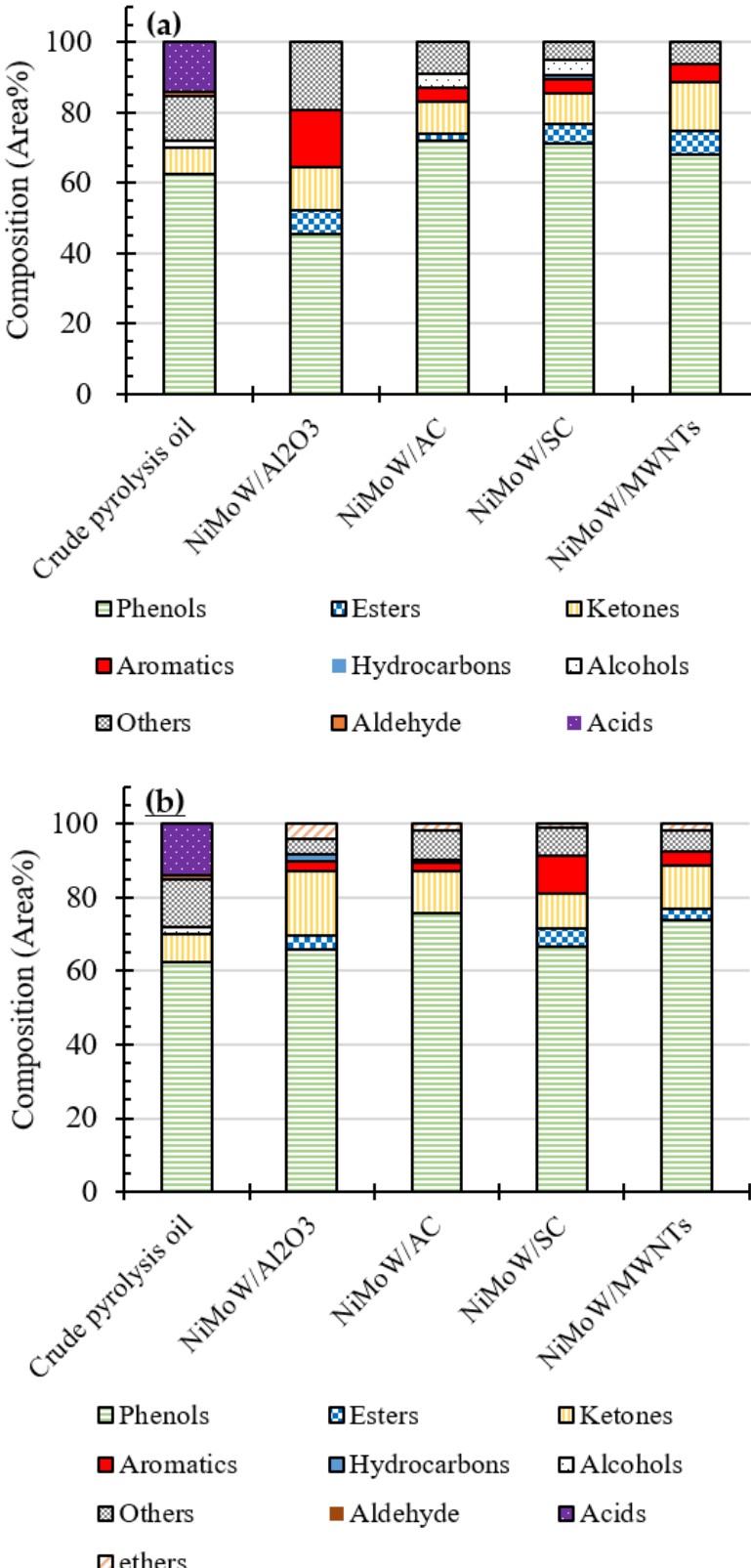

**Figure 4.** *Cont.*

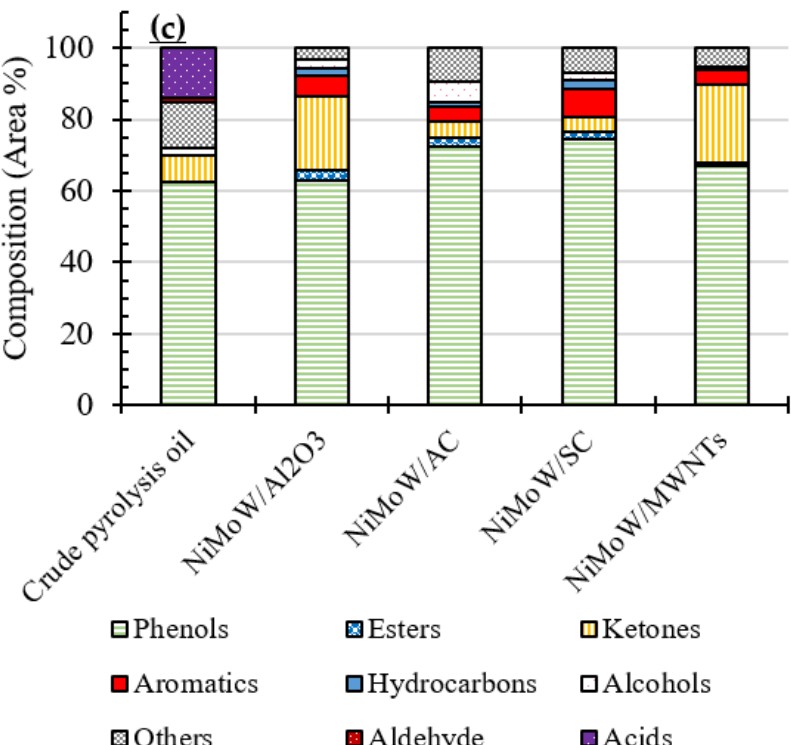

**Figure 4.** Composition distribution of the pyrolysis oil and upgraded oil obtained from HDO over NiMoW catalysts supported on different materials in supercritical ethanol with formic acid at 300 °C (**a**) or 350 °C (**b**), or with hydrogen gas at 350 °C (**c**).

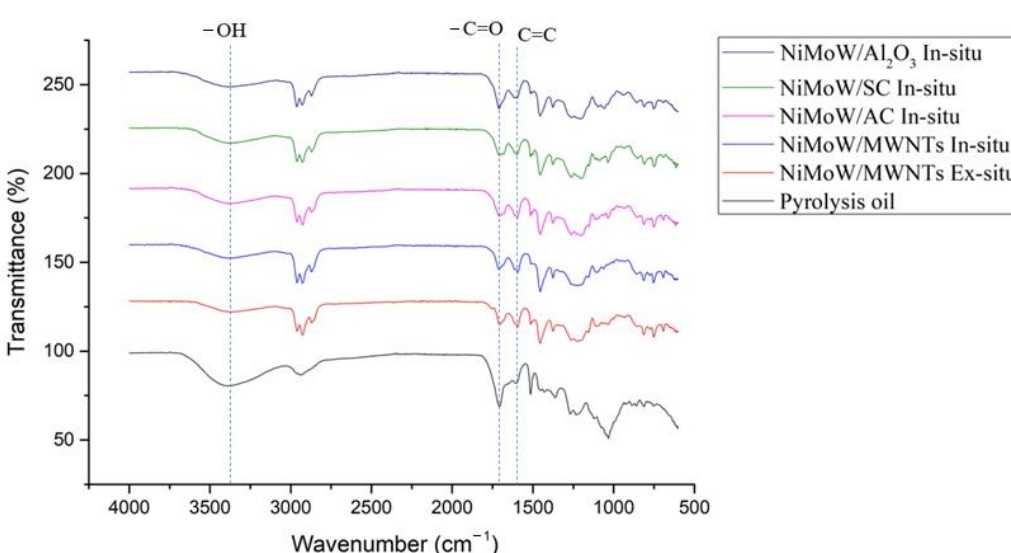

**Figure 5.** FT-IR spectra of the pyrolysis oil and upgraded oils obtained from HDO over different catalysts in supercritical ethanol with in situ hydrogen source and over NiMoW/MWNTs with ex situ hydrogen source at 350 °C.

## 4. Conclusions

In this study, a series of NiMoW catalysts supported on different materials (Al$_2$O$_3$, AC, SC, and MWNTs) were investigated for the HDO of pyrolysis oil in supercritical ethanol with in situ (formic acid) or ex situ (H$_2$ gas) hydrogen sources at 300 °C or 350 °C. Considering the difference in the metal loading, the comparable upgrading performance showed that carbon-supported catalysts are promising catalysts for the HDO of pyrolysis oil. In terms of product yields and the upgraded oil properties, the optimal reaction

condition was at 350 °C with the in situ hydrogen source over the NiMoW/MWNT catalyst. The NiMoW/MWNT catalyst showed the best HDO performance among all the catalysts tested, regardless of the hydrogen source used, likely owing to the unique pore structure of the MWNT-supported material, while the in situ hydrogen source outperformed the ex situ hydrogen source in terms of upgraded oil yields and qualities, regardless of the catalysts employed.

**Supplementary Materials:** The following supporting information can be downloaded at: https://www.mdpi.com/article/10.3390/su15107768/s1, Figure S1: Nitrogen adsorption/desorption isotherms of (a) NiMoW/Al$_2$O$_3$, (b) NiMoW/AC, (c) NiMoW/SC, (d) NiMoW/MWNTs; Table S1: Organic composition of crude pyrolysis oil; Table S2: Organic composition of upgraded oil obtained over NiMoW/Al$_2$O$_3$ at 300 °C with in-situ hydrogen source; Table S3: Organic composition of upgraded oil obtained over NiMoW/AC at 300 °C with in-situ hydrogen source; Table S4: Organic composition of upgraded oil obtained over NiMoW/SC at 300 °C with in-situ hydrogen source; Table S5: Organic composition of upgraded oil obtained over NiMoW/MWNTs at 300°C with in-situ hydrogen source; Table S6: Organic composition of upgraded oil obtained over NiMoW/Al$_2$O$_3$ at 350 °C with in-situ hydrogen source; Table S7: Organic composition of upgraded oil obtained over NiMoW/AC at 350 °C with in-situ hydrogen source; Table S8: Organic composition of upgraded oil obtained over NiMoW/SC at 350 °C with in-situ hydrogen source; Table S9: Organic composition of upgraded oil obtained over NiMoW/MWNTs at 350 °C with in-situ hydrogen source; Table S10: Organic composition of upgraded oil obtained over NiMoW/Al$_2$O$_3$ at 350 °C with ex-situ hydrogen source; Table S11: Organic composition of upgraded oil obtained over NiMoW/AC at 350 °C with ex-situ hydrogen source; Table S12: Organic composition of upgraded oil obtained over NiMoW/SC at 350 °C with ex-situ hydrogen source; Table S13: Organic composition of upgraded oil obtained over NiMoW/MWNTs at 350 °C with ex-situ hydrogen source; Table S14: All metal contents of NiMoW catalysts supported on different materials.

**Author Contributions:** Conceptualization, C.C.X. and Y.Z.; methodology, X.H.; software, M.Z. and H.W.; validation, M.Z.; formal analysis, M.Z. and H.W.; investigation, M.Z.; resources, C.C.X. and Y.Z.; data curation, M.Z. and X.H.; writing—original draft preparation, M.Z.; writing—review and editing, X.H., C.C.X., and Y.Z.; visualization, M.Z.; supervision, C.C.X. and Y.Z.; project administration, C.C.X. and Y.Z.; funding acquisition, C.C.X. and Y.Z. All authors have read and agreed to the published version of the manuscript.

**Funding:** This study was funded by Natural Resources Canada (NRCan) Forest Innovation and OERD Clean Energy programs, Mitacs/Western Maple Bio Resources Inc. Accelerate Internship (IT18350), as well as the Discovery Grant (RGPIN-2019-05159) from the Natural Science and Engineering Research Council of Canada (NSERC).

**Institutional Review Board Statement:** Not applicable.

**Informed Consent Statement:** Not applicable.

**Data Availability Statement:** The data presented in this study are available upon request from the corresponding author.

**Acknowledgments:** The authors gratefully acknowledge the technical support from NRCan CanmetMATERIALS, the Institute for Chemicals and Fuels from Alternative Resources, and the Surface Science Western at Western University.

**Conflicts of Interest:** The authors declare that they have no known competing financial interests or personal relationships that could have appeared to influence the work reported in this paper.

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
