# Peer review of "Hydrodeoxygenation of Pyrolysis Oil in Supercritical Ethanol with Formic Acid as an In Situ Hydrogen Source over NiMoW Catalysts Supported on Different Materials"

_sustainability, doi:10.3390/su15107768_

Round 1
Reviewer 1 Report
- Please define NiMoW in the abstract part.
- Since most of the time, the abstract should be ‘stand-alone’, it is great if the results can be specified into a number like a percentage.
- For methodology 2.2, please include your method references. It would be great if the “calculated amounts” could be specified.
- For methodology 2.3, please include your references. The experiments were performed in 100 or 500 mL, depending on the temperature. Does it mean 100 mL at 300°C and 500 mL at 350°C? I think the write-up can be improved for better understanding. For a 100 mL reactor, the total mass of bio-oil and ethanol is 40g, equivalent to 40% of the total reactor volume, while for a 500 mL reactor, the total mass is 80g, equivalent to 16% only. Why? It shows that the headspace volume for both reactors differs; thus, could it affect the reaction? Please also point out the different types of catalysts used in the experiments. Is the wavenumber range for the FTIR correct? Because it is different from the wavenumber range in part 3.5.
- For methodology 2.4, please include your references.
- For methodology 2.5, please include your references.
- For result part 3.1, authors should include references from previous research to support their results.
- Why were the results for ex-situ 300°C not included?
- In section 3.3, where do the initial or comparing values of DOD and oil properties come from? Is it from your result or previous research reports? Please specify and include references. If it is your results, it would be great to include them in the table.
- For the FTIR result, why the other catalyst is not included?
- Please highlight the novelty of this research.
- The turn-it-in software shows that the plagiarism index is 36%. Please do something to reduce it to below 20%.

Reviewer 2 Report
The upgrading pyrolysis oils (from wood) was carried out by HDO process in supercritical ethanol using formic acid as an in-situ hydrogen source (at 300 °C and 350°C). The hydrogen gas (ex-situ hydrogen source) was also used for comparison over NiMoW catalysts supported at Al2O3, activated carbon, sawdust carbon, and multiple walls nanotubes (MWNTs). A series of issues need to be better clarified to improve the quality of the work. The authors do not discuss the actual active phase of the catalyst. Furthermore, no test without a catalyst was performed to verify the thermal effect of the reaction either in the presence of formic acid or not, considering that ethanol is a hydrogen donor. Thus, only by major revision, this paper should be reconsidered for publication.
Please consider many remarks:
1- Pag. 78: “The crude bio-oil used in this work is a commercially available fast pyrolysis oil derived from woody biomass”. Which company supplied this bio-oil
2- Pinewood sawdust is used as the precursor for sawdust carbon (SC), however the preparation procedure for the support is unclear, was the material first impregnated and then carbonized? Is there a reason why the support wasn’t carbonized before impregnation and then calcined?
3- About the impregnation of nickel, molybdenum, or tungsten how were they impregnated? By which method? Successive impregnations?
4- The objective was to prepare catalysts with 10, 6, and 5% of Ni, Mo, and W? How to explain the values obtained by ICP-AES? The W was not quantified in these catalysts?
5- The amount of catalysts and formic acid was loaded in relation to the mass of bio-oil on a dry basis, so considering 20 g of bio-oil, what was the mass of catalyst and mass of formic acid used, respectively?
6- For analysis of liquid bio-oil, was the liquid extracted from the evaporation step injected directly into the GC-MS? Did you dilute it with some solvent?
7- About the HDO process: were the catalysts previously reduced?
8- About the XRD analysis (FIG. 1): Specify the powder diffraction files identified in each catalyst (PDF Data Card). Also, for the NiMoW/AC catalyst, the diffraction line at 29° refers to what?
9- Please replace the term spectrum with diffractogram line 191.
10- Could you explain how the catalyst NiMoW/MWNTs showed the diffraction line of MoO2 after calcination in an inert atmosphere (N2 flow at 400 °C for 3 h)? And about the other catalysts also supported at carbon?
11- The authors claim that the good performance of NiMoW/MWNTs can be attributed to its pore structure, which improves the access of the bio-oil molecules to the active sites of the catalyst, however, no discussion is held about the active sites for the HDO reaction. In this case (NiMoW catalysts), what are the active sites for the HDO reaction?
12- I did not understand the statement of the sentence page 217: “ The reasons for the significantly increased gas yield at a high temperature could be due to formic acid decomposition and enhanced cracking reactions at a higher temperature [36].”
What do the authors mean by formic acid decomposition? Shouldn't it already decompose at a temperature much lower than 300 °C?
13- On Figure 4 the Composition of oils is separated in phenols, ketones, and acids… Could you please a table with all identified compounds (perhaps in support information)?
14- No blanks were performed? HDO test without catalyst with or without formic acid? What is the effect of temperature and solvent?
1- The amount of catalysts and formic acid was loaded in relation to the mass of bio-oil on a dry basis, so considering 20 g of bio-oil, what was the mass of catalyst and mass of formic acid used, respectively?
2- For analysis of liquid bio-oil, was the liquid extracted from the evaporation step injected directly into the GC-MS? Did you dilute it with some solvent?
3- About the HDO process: were the catalysts previously reduced?
4- About the XRD analysis (FIG. 1): Specify the powder diffraction files identified in each catalyst (PDF Data Card). Also, for the NiMoW/AC catalyst, the diffraction line at 29° refers to what?
5- Please replace the term spectrum with diffractogram line 191.
6- Could you explain how the catalyst NiMoW/MWNTs showed the diffraction line of MoO2 after calcination in an inert atmosphere (N2 flow at 400 °C for 3 h)? And about the other catalysts also supported at carbon?
7- The authors claim that the good performance of NiMoW/MWNTs can be attributed to its pore structure, which improves the access of the bio-oil molecules to the active sites of the catalyst, however, no discussion is held about the active sites for the HDO reaction. In this case (NiMoW catalysts), what are the active sites for the HDO reaction?
8- I did not understand the statement of the sentence page 217: “ The reasons for the significantly increased gas yield at a high temperature could be due to formic acid decomposition and enhanced cracking reactions at a higher temperature [36].”
What do the authors mean by formic acid decomposition? Shouldn't it already decompose at a temperature much lower than 300 °C?
9- On Figure 4 the Composition of oils is separated in phenols, ketones, and acids… Could you please a table with all identified compounds (perhaps in support information)?
10- No blanks were performed? HDO test without catalyst with or without formic acid? What is the effect of temperature and solvent?

Reviewer 3 Report
-
The authors present a topic about hydrodeoxygenation of pyrolysis oil using NiMoW catalysts supported on different support materials (including Al2O3, activated carbon, sawdust carbon, and multiple walls nanotubes) under supercritical ethanol with formic acid as an in-situ hydrogen source. The HDO product oil analysis data is quite good. However, W is not found in catalyst using carbon-based support materials. And authors use the different metal contents in different catalysts. Thus, the arguments and discussion of findings are not compelling.
It is suggested that the authors should improve the following aspects to form a publishable manuscript.
- In the introduction part, “Inexpensive catalysts, such as conventional hydrotreatment or petroleum refinery catalysts, including NiMoW, NiMo and CoMo sulfides supported on different materials, have been applied for bio-oil catalytic HDO upgrading.” was written. Why did author focus on the NiMoW catalysts? Please mention it in the introduction parts.
- Authors mentioned about the noble catalysts applied for HDO treatment with high performance. This manuscript gives the main discussion on the product oil properties. However, the information of HDO product oil was not mentioned in the case of noble catalysts. Thus, it is different to understand the different behaviors between two system catalysts. Please add the performance of noble catalyst in more detail in the introduction part.
- A commercial catalyst NiMoW/Al2O3 was employed; and other catalysts were prepared at laboratory with different metal contents and different supports. How can author compare the performances and give the conclusion with the impropriate samples? Please give the more results of addition samples and discuss about the relationship between pore structure and performances.
- I would like to suggest that the author should check the manuscript carefully to correct the grammar, mistyping words, and enhance the soundness of scientific writing.
Round 2
Reviewer 2 Report
Many issues are not clear and need to be better clarified. For example, regarding the blank test, the high N content and low oxygen content. Why was the DOD% formula modified? It would be interesting to present the values of all metals in the catalysts. The text of question 3 was not completely written in the manuscript.
Reviewer 3 Report
Thank authors for carefully modifying the manuscript. However, It is suggested that the authors should improve the following aspects to form a publishable manuscript.
1. In the introduction part, " In a study of Oh et. al [25], Ru/C and Pt/C were used in hydrodeoxygenation of bio-oil at 250, 300, and 350 °C, it was found that the degree of deoxygenation can reach to 73.7 over Pt/C catalyst at 350 °C." was added. What is unit of 73.7? Please re-write the sentence to be clear statement.
2. In the Results and Discussion section, "NiMoW/SC has lower Ni and Mo contents comparing with other catalysts, which may be caused by the low surface area and pore volume." was written.
Ni and Mo tend to agglomeration due to low surface area and pore volume, which leads to metal lost during catalyst synthesis (impregnation => drying => calcination).
Please re-write the sentence.
3. "Besides, Co and Mo contents are lower than target loading" was written. Please change Co to Ni.
4. "This suggests that both the large specific area and mesoporous structure of the catlysts contribute to their superior catalytic performance." was written. However, please reconsider the effect of pore volume and pore diameter factor on the bio-oil diffusion and active site accessability to make more concrete conclusion.
Round 3
Reviewer 2 Report
Accept in present form